# Bacterial Periodontitis in Horses: An Epidemiological Study in Southern Italy

**DOI:** 10.3390/ani13111814

**Published:** 2023-05-30

**Authors:** Leonardo Occhiogrosso, Paolo Capozza, Alessio Buonavoglia, Nicola Decaro, Adriana Trotta, Claudia Marin, Marialaura Corrente

**Affiliations:** 1Department of Veterinary Medicine, University of Bari Aldo Moro, 70010 Valenzano, Italy; 2Department of Education, Psychology, Communication, University of Bari, 70121 Bari, Italy

**Keywords:** horse, periodontitis, red complex bacteria

## Abstract

**Simple Summary:**

Equine periodontal disease (EPD) is a painful syndrome in horses that causes significant health problems. Despite its clinical importance, very few investigations on its etiopathogenesis have been performed. This study investigates the association of different bacterial species, including the red complex bacteria (RCB, i.e., *Treponema denticola*, *Tannerella forsythia* and *Porphyromonas gingivalis*), *Fusobacterium nucleatum*, *Veilonella parvula*, and *Prevotella intermedia*, with periodontal disease and other risk factors. *Tannerella* spp. and *Treponema* spp. were detected with a significantly higher proportion in horses affected by EPD than in healthy animals. Overall, the investigated pathogens, belonging to RCB, were detected in a low number of horses. Age was found to be the main risk factor, with horses aged older than 20 years being at higher risk of EPD. A very high rate of coinfection was statistically associated with EPD, and, thus, EPD supported the multiagent etiology of this equine complex syndrome.

**Abstract:**

Equine periodontal disease (EPD) is a painful oral inflammatory syndrome characterized by multifactorial pathogenesis. Although it is well known that bacterial proliferation and consequent gingivitis are caused by the decomposition process of food residues, in hypsodont species, the pathogenetic role and the different bacterial species involved in the progression of EPD must be fully clarified. This study aimed to investigate the association of bacteria, including the complex red bacteria (RCB), with EPD, and to evaluate possible EPD risk factors. Bacterial species, including *Treponema denticola*, *Tannerella forsythia*, *Porphyromonas gingivalis* (belonging to the RCB), *Fusobacterium nucleatum*, *Veilonella parvula*, and *Prevotella intermedia*, were investigated in 125 oral swabs from healthy and EPD-affected horses using real-time multiplex PCR. Subsequently, possible risk factors (i.e., age, gender, and breed of the animals and type of feed used) were evaluated using univariate and multivariate analyses. *Tannerella* spp. and *Treponema* spp. were detected in a significantly higher proportion of horses affected by EPD than in healthy animals, although pathogens belonging to RCB were detected in low number of horses. At the same time, none of the investigated pathogens was significantly associated with a particular stage of disease severity. Horses aged older than 20 years were at higher risk of EPD. The high rate of coinfection, statistically associated with EPD, supports the hypothesis that EPD is a complex syndrome characterized by the possible simultaneous involvement of several pathogens and an increased risk depending on the animal’s age. Constant oral hygiene is the best prevention to prevent and treat the disease, especially in old animals.

## 1. Introduction

Periodontal diseases (PDs), such as gingivitis and periodontitis, are multifactorial inflammatory syndromes of the mouth [1,2,3]. PDs are common in humans [4] and in other animal species, such as dogs [3,5], donkeys [6,7], and sheep [8]. In horses, the disease is recognized as a painful condition, which poses significant health problems [9,10]. Gingivitis designates inflammatory lesions confined to the marginal gingiva and is characterized by bleeding and inflammation. It is generally accepted that some cases of gingivitis can evolve to periodontitis. Periodontitis occurs when lesions have extended beyond the gingiva to include destruction of the structures underlying the connective tissue that support the teeth, damaging the bone and periodontal ligament and leading to tooth loss [9,10,11,12]. Periodontitis is a complex disease process involving genetic, microbial, immunological, and environmental factors that determine the risk of acquiring and progression of the disease [11,12].

Although the first observation of periodontal syndrome in equids dates to 333 BC by Aristotle [13], the main clinical features of PDs in the domestic horse were not described until 1900. In 1906, Colyer first reported PD as the “scourge of the horse”, finding this disease in approximately 33.5% (162/484) animals examined [14]. In the 1930s, PD was found in 30% of 50 equine skulls examined [15]. In that study, it was concluded that the disease was initiated by gingival trauma caused by the feeding of coarse chaff [15]. Subsequent investigations in the 1970s and 1990s in the UK and USA have shown similar high prevalence levels of the disease [16,17]. It was found that, in an examination of the teeth and gums of 218 and 446 skulls, respectively, the incidence of periodontal disease changed with age. There was a 40% prevalence in horses of three to five years of age; the prevalence decreased in horses aged between five and ten years and then increased to 60% in horses over the age of 15 years [16,17]. Recent studies confirmed a variable prevalence of the disease, ranging from 10% to 75%, depending on various factors, including the country, age, sex, and breed of the animals, as well as the type of feed administered [17,18]. While PD is reported worldwide in the equine population, information on the prevalence of the disease in Italian domestic horses is still missing [17,18,19,20,21,22,23,24,25]. There is evidence that, in species with brachydont teeth, such as humans and domestic carnivores, PD is linked to dysbiosis and determined by the accumulation of dental plaque in the gingival sulcus [26,27]. Such factors may alter the tooth surface and favour the attachment and replication of bacteria in mixed growths [28]. A similar relationship exists in canine [3,5,29] and feline [30] PD.

The recent keystone pathogen hypothesis proposes that specific pathogens present in the oral cavity, albeit in low abundance, may modulate their environment, disrupting the ordinarily symbiotic relationship between the bacteria of the oral cavity, resulting in dysbiosis (dysregulation of commensal oral bacteria), and thereby contributing to the development of inflammatory disease [9].

In humans, three microorganisms are mainly associated with PD, i.e., *Treponema denticola*, *Tannerella forsythia*, and *Porphyromonas gingivalis*, which are usually referred to as forming the so-called red complex bacteria (RCB) [4,28]. These bacteria are Gram negative, non-spore-forming anaerobic organisms, and they may be found as pure or mixed infections [4,28]. RCB possess several virulence factors, including fimbriae, proteinases, exopolysaccharides, and hemin-binding proteins [31]. RCBs have been detected both in subgingival plaque and in the apical root canal, causing periodontal and endodontic diseases [28,32,33]. *Porphyromonas gingivalis* is a well-known keystone pathogen in human periodontal disease due to its ability to modulate the host immune system, thus altering host immune responses to the entire oral biofilm [9,34].

Although plaque and calculus are observed in teeth of hypsodont species, as equines, they rarely lead to attachment loss. In most cases of equine periodontal disease, no calculus or plaque is present. Equine periodontal disease (EPD) has historically been thought to be almost entirely a secondary disease process. Primum movens of EPD is represented by the stasis and decomposition of food residues in the small depressions around and between the teeth [14,35], both caused by physical disorders of tooth growth, eruption, and wear [9,17,22,36]. The decomposition process of food residues causes gingival inflammation and bacterial proliferation, which can lead to the development of EPD [9,37,38].

Reports about the equine oral microbiome and the role of bacteria in the etiology of EPD are scarce [9,10,37,38]. In addition to RCB, other bacterial species, including *Fusobacterium nucleatum*, *Veilonella parvula*, and *Prevotella intermedia*, have been proposed as potential etiological agents of EPD [9,37,38].

Studies in other species have estimated that around 50% of oral bacteria cannot be cultivable in vitro or easily distinguished in culture [39]. Polymerase chain reaction (PCR) offers a highly sensitive and specific tool for detecting bacteria without cross-reactivity with other oral microorganisms [40].

In this study, the presence of RCB and other bacteria, such as *Fusobacterium nucleatum*, *Veilonella parvula*, and *Prevotella intermedia*, was investigated by using PCR protocols in oral mucosal swabs from horses affected by EPD and in healthy horses. Furthermore, the association between the severity of lesions due to EPD and different risk factors was evaluated.

## 2. Materials and Methods

### 2.1. Sample Collection

A total of 125 oral mucosal swabs were collected in the period between April 2020 and April 2021. Samples were collected from different regions in Italy as follows: Apulia (*n* = 112), Campania (*n* = 6), Basilicata (*n* = 2), and Molise (*n* = 5). Sixty-five oral swabs (collection A) were taken from horses affected by EPD, and a total of 60 oral swabs (collection B) were collected from healthy animals. Data regarding the region where the animals were housed, age, sex, breed, and type of feed administered to the horses were collected. Health status was assessed by general physical examination, using a body condition score (BCS). The BCS was determined on the basis of the Henneke body condition scoring system guidelines for horses [41]. Furthermore, an accurate oral examination was conducted on the animals after the application of a dental speculum (Apribocca mod. Millennium, Dearson, Birmingham, UK) and after intravenous sedation (10–30 µg/Kg of detomidine hydrochloride). The mouths of all examined individuals were rinsed with warm water and a pressure lance (Dearson, Birmingham, UK) to remove food debris from the interdental spaces. Complete inspection of the mouth was performed with a LED head light and a dental mirror (Dearson, Birmingham, UK), while the depth of periodontal pockets present was determined with a periodontal probe (Kruuse, UK). DNA/DNase/RNase-free swabs (Cliniswab, APTACA, Asti, Italy) were used to collect the oral material found on the surface of healthy teeth (cheek teeth) and, in diseased subjects, in the gingival pockets, dental fractures, wounds, and/or diastemata (Figure 1). None of the horses screened had a regular oral routine. The samples were stored at room temperature and sent the same day to the laboratories of the section of Infectious Diseases, Department of Veterinary Medicine (DiMeV) at the University of Bari, where they were kept at +4 °C until processing within 12 h.

#### 2.1.1. DNA Extraction from Oral Swabs

Dry swab samples were homogenized in 10% Dulbecco’s modified Eagle’s medium (DMEM) and then centrifuged at 10^4^× *g* 3 min. Nucleic acids were extracted from 200 µL of the supernatants using the QIAamp Cador Pathogen Minikit (Qiagen S.p.A., Milan, Italy), following the manufacturer’s protocol and stored at −80 °C until use. DNA eluted in 100 μL of RNase-free water was quantified using the spectrophotometer model NanoDrop ND—1000 V3.8 (TermoFisher, Waltham, MA, USA).

#### 2.1.2. Multiplex Real-Time PCR

Two multiplex real-time PCRs (qPCRs) were set up. Primers and probes previously described [42,43,44] (Table 1 and Figure 1) were grouped for the different pathogens to be tested. Specifically, the multiplex qPCRs were set up as follows: (i) PAROD 1 aimed at identifying *Fusobacterium nucleatum*, *Treponema denticola*, and *Tannarella forsythia*; (ii) PAROD 2 aimed at identifying *Prevotella intermedia*, *Prevotella nigrescens*, and *Veilonella parvula*. The mixture was prepared in a total volume of 12.5 μL containing 1 μL iTaqTM Universal Probes Supermix (Bio-Rad Laboratories Srl, Milan, Italy), 10 μL c-DNA, 600 nM of each primer, and 200 nM of each probe (Table 1).

Amplification reactions were performed on a real-time system thermocycler, CFX96TM (Bio-Rad Laboratories Srl, Milan, Italy). The thermal protocol consisted of activation of iTaq DNA polymerase at 95 °C × 2 min., followed by 45 cycles of denaturation at 95 °C × 15 s, annealing at 55 °C × 30 s, and extension at 60 °C × 1 min. The fluorescent signal was detected during the reaction’s extension phase, and the data were analyzed using the appropriate sequence detection software (Bio-Rad CFX Manager v. 3.1, Bio-Rad Laboratories Srl). To verify the absence of DNA loss during the extraction phase and the presence of PCR inhibitors, an internal positive control (IC), consisting of ovine herpesvirus type 2 (OvHV-2), was added to the lysis buffer (AVL buffer, QIAGEN S.p.A.) before nucleic acid extraction. The fixed amount of IC added to each sample was calculated so that the average cycle threshold (C*_T_*) value in the specific qPCR assay was 34.18 with a SD of 0.65, calculated from 50 separate runs. Samples with a C*_T_* value > 35.48 (average plus two standard deviations) were excluded from the analysis.

#### 2.1.3. Data Analysis

Statistical analysis of variables was performed using R software version 4.0.2 (R Foundation for Statistical Computing, Vienna, Austria; https://www.R-project.org/ (accessed on 23 May 2022)). Categorical data were summarized as count and percentage. Fisher’s exact test and chi-square test were used to analyze categorical variables, when appropriate. The measure of association between qualitative variables (type of lesion and presence of pathogens) was determined using Cramer’s Index V, which was calculated according to previously published guidelines [45]. Univariate analysis was performed to identify categorical variables, including region of animal origin, age (young [1 ≤ age ≤ 10] vs. adult [11 ≤ age ≤ 20]; young [1 ≤ age ≤ 10] vs. old [age > 20]), sex, breed, and overall physical condition, using BCS (BCS = 5 versus. 1 ≤ BCS ≤ 4 and BCS = 5 vs. 6 ≤ BCS ≤ 9).

The EPD evaluated at the oral examination was classified into different categories (mild gingivitis, diastema, fracture, pocket, and others) according to the severity of the disease, as previously suggested [9,46].

To evaluate whether observed variables could be considered as useful predictors of the presence of the disease, a logistic regression analysis (multivariate) was performed. The clinical variable sick (1) or healthy (0) was assigned based on a complete examination of the oral cavity of the horses. The predictors were the following variables: positivity to bacteria (*Treponema denticola*, *Tannerella forsythia* and *Porphyromonas gingivalis*, *Fusobacterium nucleatum*, *Veilonella parvula*, and *Prevotella intermedia*), coinfection, age, sex, type of food administered to the horses, breed, and region of origin of the animals. The magnitude of the association between the variables and the presence of bacteria was expressed as odds ratio (OR) with 95% confidence intervals (95% CI). A *p*-value < 0.05 was considered statistically significant.

## 3. Results

A total of 125 animals were included in the study, and the categorical data (animal origin, age, sex, type of feed administered, and breed) were recorded for each sample. Seventy-seven males and 48 females were included in the study, with ages ranging from three to thirty-two years (mean 14.5 years, median 14 years, 95% CI 13.2–15.8). Of these, 32.8% (41/125) were less than 10 years old (young horse), 46.4% (58/125) were between 11 and 20 years old (adult horse), and 20.8% (26/125) were older than 20 years (old horse). Most animals were American Saddlebred horses (79.2%, 99/125), lived in the Apulia region (89.6%, 112/125), and were fed with hay and concentrates (97.6%, 122/125). The BCS reported for each animal included in the study ranged from 1 (poor) to 7 (fleshy). Of these, 78.4% (98/125) of the horses had a BCS of 5 (moderate). On clinical oral evaluation, 52% (65/125) of the animals were found to be affected by EPD. Of these, 52.3% (34/65) were affected by periodontal pockets, 26.1% (17/65) had diastema, 7.6% (5/65) were affected by fractures, and 13.8% (9/65) had others lesions attributable to mild EPD. The prevalence of RCB (*Treponema denticola*, *Tannerella forsythia*, and *Porphyromonas gingivalis*), *Fusobacterium nucleatum*, *Veilonella parvula*, and *Prevotella intermedia*, using PCR protocols, in association with a different category (mild gingivitis, diastema, fracture, periodontal pocket, and others), based on the severity of EPD, is reported in Table 2. No pathogen was statistically correlated with any particular type of lesion (Table 2).

EPD associated with region of animal origin, age, sex, type of feed administered, breed, as well as bacterial positivity, is shown in Table 3.

Overall, 91.2% (114/125) animals tested positive for *Veilonella parvula*, 60.8% (76/125) tested positive for *Fusobacterium nucleatum*, 54.4% (68/125) tested positive for *Tannerella forsythia*, 44.8% (56/125) tested positive for *Treponema denticola*, 29.6% (37/125) tested positive for *Prevotella intermedia*, and 2.4% (3/125) tested positive for *Porphyromonas gingivalis*. To evaluate the causative role of the bacteria, the PCR results were statistically analyzed, comparing the groups, i.e., animals with EPD (collection A; *n* = 65) and healthy horses (collection B; *n* = 60). Of the horses with EPD (collection A), 61.5% (40/65) tested positive for *Treponema denticola*, 78.4% (51/65) tested positive for *Tannerella forsythia*, 67.6% (44/65) tested positive for *Fusobacterium nucleatum*, 3% (2/65) tested positive for *Porphyromonas gingivalis*, 33.8% (22/65) tested positive for *Prevotella intermedia*, and 95% (62/65) tested positive for *Veilonella parvula*. Of 60 healthy horses (collection B), 26.6% (16/60) were positive for *Treponema denticola*, 28.3% (17/60) were positive for *Tannerella forsythia*, 53.3% (32/60) were positive for *Fusobacterium nucleatum*, 1.6% (1/60) were positive for *Porphyromonas gingivalis*, 25% (15/60) were positive for *Prevotella intermedia*, and 88.3% (53/60) were positive for *Veilonella parvula*.

The results of multivariate analysis are shown in Table 4. The risk of EPD in horses was significantly associated with animal age and positivity for *Treponema denticola* and *Tannerella forsythia* in the univariate analysis and multivariate model (Table 3 and Table 4).

Horses older than 20 years were found to have a higher risk of developing EPD compared to young horses, both in univariate (OR = 5.67; 95% IC: 1.75–20.36; *p*-value = 0.001211) and multivariate analysis (OR = 7.98; 95% IC: 2.16–29.5; *p*-value = 0.00186). According to the univariate analysis, adult animals aged 11–20 years have about three times higher risk of exhibiting EPD than young horses (OR = 2.81; 95% CI: 1.14–7.21; *p*-value = 0.02316).

*Treponema denticola* and *Tannerella forsythia* were found to be significantly associated with the development of the disease (*p*-value < 0.05), both using the univariate (OR = 4.34; 95% IC: 1.93–10.13; *p*-value = 0.000137 and OR = 6.49; 95% IC: 2.82–15.63; *p*-value = 0.000001015, respectively) and multivariate analysis (OR = 3.24; 95% IC: 1.15–9.15; *p*-value = 0.02620 and OR = 4.7; 95% IC: 1.79–12.4; *p*-value = 0.00172, respectively). Cramer’s V coefficient indicates average intensity of the relationship (V = 0.475 and V = 0.451 respectively).

No association was observed for the other variables (i.e., origin of the animal, sex, reproductive status, type of feed fed to the horses, BCS, and breed) potentially related to EPD (Table 4).

Furthermore, 14.4% (18/125) of horses tested positive for five pathogens, 18.4% (23/125) of horses tested positive for four pathogens, 23.2% (29/125) of horses tested positive for three pathogens, 20% (25/125) of subjects tested positive for two pathogens, and 18.4% (23/125) of horses tested positive for only one pathogen. Finally, only 1.6% (2/125) of subjects tested positive for all six pathogens combined.

Animals who were positive for four pathogens simultaneously were almost ten times more likely to develop EPD than patients who are positive for only one pathogen. Likewise, patients who are positive for five pathogens, simultaneously, have a six-fold higher risk of evolving EPD than those who are positive tor a single pathogen.

## 4. Discussion

EPD is a painful multifactorial inflammatory syndrome of the mouth and highly prevalent disorder in horses that causes a significant health problem [9].

The prevalence of EPD in the equine population varies widely and generally depends on the type of population studied, the collection of samples intra vitam or post mortem (upon examination), the number of subjects included in the study, and the country where the study was conducted [47]. In this study, more than 50% of the animals examined were affected by EPD, displaying different degrees of severity. The reported prevalence of EPD varies from 6.1% in Sweden to more than 60% in Australia [18,22,48]. However, comparing these reports is very difficult because most of the studies were conducted on post mortem samples, and very few investigations were performed on lived horses. In particular, a UK study reported an EPD prevalence of 49.9% [23]. These very high prevalence rates were subsequently confirmed by another epidemiological study in the UK, reporting a prevalence of 57.1% in 706 horses [49]. Finally, researchers in Western Australia found an EPD prevalence of 58.8% in 500 horses [24].

Based on our knowledge of the literature, there are a plethora of risk factors for the development of EPD, including age, sex, and breed of the animals, as well as the type of feed used [24]. In our study, a slightly higher prevalence of EPD was found in male animals, but this difference was not statistically significant. Previous studies have reported a higher incidence of dental fractures in males than in females, as they appear to exhibit more aggressive behavior toward same-sex subjects of their species when allowed to display their natural behavioral patterns [24,25]. However, the animals enrolled in the study were generally kept in individual stalls and were rarely released in the paddocks, a factor that probably prevented the occurrence of aggression and, consequently, oral injuries in the males.

The correlation between animal age and onset of ECP are supported by other studies [25,50,51]. Adult (10–20 years) and older (>20 years) horses had approximately three- and six-fold higher risks, respectively, than young horses (<10 years). An increase in the likelihood of disease was observed with increasing age, an aspect also seen in other species, such as donkeys [7], dogs, and humans [4,5]. In all these species, an excellent oral routine can prevent PDs and is an optimal strategy to reduce the extent of the disease in old age and the associated pain symptoms.

Interestingly, our study revealed a higher percentage of subjects presenting with an advanced stage of periodontal syndrome (diastema/pocket/fracture), as reported in previous studies [18,23]. Although EPD is common in the equine population [20], it is often underdiagnosed in early phases because horses have a naturally high pain tolerance threshold [52]. When horses experience teeth pain, they seem to be able to change their chewing pattern by using the natural dental arch unilaterally, even for years. Owners do not always notice this change in behavior [53]. This condition could favor the accumulation of food between the teeth with the formation of periodontal pockets, diastema, and the subsequent pathogenetic development of the oral disorder [9].

EPD differs from plaque-induced periodontitis in humans [4], dogs [3,5,29], and cats [30], where bacteria accumulating in dental plaque induce a destructive inflammatory response in the periodontium. In horses, the leading cause of EPD is the stasis and decomposition of entrapment of feed residue between cheek teeth [14,35], causing inflammation of periodontal tissue and bacterial infection of the periodontal tissues, further exacerbated by the host’s response [9].

In the present study, *Tannerella* spp. and *Treponema* spp. were detected in a significantly higher proportion of horses affected by EPD, although the pathogens belonging to RCB were detected only at low levels. At the same time, none of the pathogens investigated was significantly associated with any particular level of disease severity.

*Treponema denticola* and *Tannerella forsythia* are recognized as periodontal pathogens in humans, acting as components of RCB found in severe periodontitis lesions alongside *Porphyromonas gingivalis* [9,10,38]. In a study conducted in Austria using PCR assays, *Tannerella* spp. and *Treponema* spp. were more commonly isolated from horses with periodontitis secondary to equine odontoclastic tooth resorption and hypercementosis (EOTRH) than in healthy horses [38]. More recently, in a study examining the microbiome of the equine oral cavity associated with EPD, using a high-throughput sequencing approach, abundance of both genera was found to be significantly increased in the periodontitis [10].

Another pathogen strictly associated with EPD is *Veilonella parvula* [9]. Although it was the bacterium most frequently detected in our samples, the differences between animals with EPD and healthy animals were not statistically significant.

Interestingly, our study revealed very high coinfection rates in horses with EPD. Patients positive for two pathogens had a seven-fold higher likelihood to develop the disease, whereas patients positive for three pathogens were six- to nine-fold more likely to develop the disease, and, when the pathogen *Tannerella forsythia* was present, they were two-fold more likely to develop the disease than when the pathogen *Fusobacterium nucleatum* was present, supporting the hypothesis of synergism as the most likely risk factor in causing periodontal pathology, rather than the presence of a single pathogenic species [4,5,9,10,38].

Some opportunistic pathogenic species may benefit from synergism, as already reported in studies on the canine microbiota, which found a positive statistical association between the presence of *Tannerella forsythia* and coinfection with *Porphyromonas gingivalis* and gingivitis/periodontitis in dogs [5,29]. Indeed, *Tannerella forsythia* may benefit from the presence of other species, so that the inflammatory process is associated with a synergistic and consequent effect as a causal effect of periodontitis. In vivo studies with animal models have confirmed the increased risk of disease development when *Tannerella forsythia* cooperates with *Fusobacterium nucleatum*, determining inflammatory responses and oral biofilm development [54]. Thus, EPD can be considered as a complex syndrome characterized by the possible involvement of multiple pathogens that act preferentially in synergy in the development of periodontal inflammation, but whose role in the pathogenesis of the oral syndrome remains to be investigated.

## 5. Conclusions

Despite the study’s potential limitations, represented by the convenience sampling and significant overrepresentation of the samples collected in Apulia, this investigation provides new information on the prevalence of EPD in the equine population in southern Italy. It examines the distribution of the main bacterial species involved in equine oral pathology while providing further information on other possible risk factors associated with the disease. EPD is a complex pathological phenomenon influenced by numerous factors, and proper oral hygiene and prevention are the most effective prophylactic methods. Further investigations are required to provide more information on the epidemiology, etiopathology, and early diagnosis of EPD. Horse owners should be better sensitized and informed about the early signs of dental pain in order to start treating oral disease as soon as possible. Horse owners should be able to detect early signs of dental pain and alert practitioners in order to start treating oral diseases as soon as possible.

## Figures and Tables

**Figure 1 animals-13-01814-f001:**
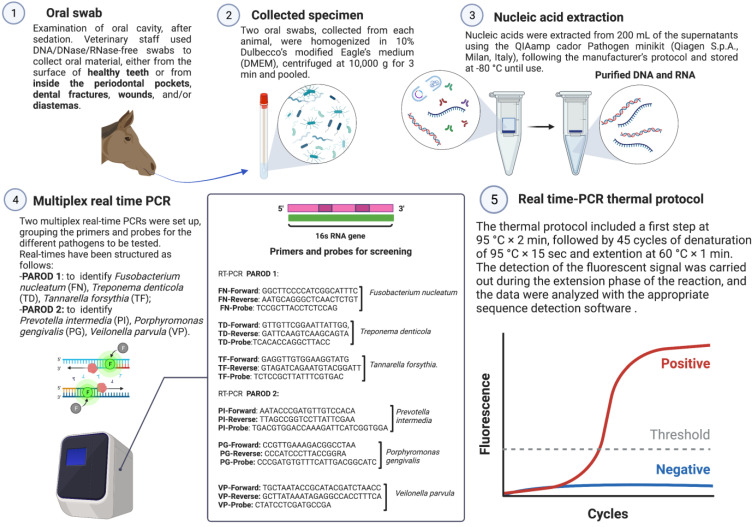
Phases of the investigation.

**Table 1 animals-13-01814-t001:** List of primers used in the study.

Specie Target	Primers and Probes		Sequence (5′-3′)	Size	References
*Fusobacterium nucleatum*	Primer	FN-For	GGCTTCCCCATCGGCATTTC	123 bp	[44]
		FN-Rev	AATGCAGGGCTCAACTCTGT		
	Probe	FN-Pb	TCCGCTTACCTCTCCAG		
*Treponema denticola*	Primer	TD-For	GTTGTTCGGAATTATTGG	109 bp	[44]
		TD-Rev	GATTCAAGTCAAGCAGTA		
	Probe	TD-Pb	TCACACCAGGCTTACC		
*Veilonella parvula*	Primer	VP-For	TGCTAATACCGCATACGATCTAACC	66 bp	[43]
		VP-Rev	GCTTATAAATAGAGGCCACCTTTCA		
	Probe	VP-Pb	CTATCCTCGATGCCGA		
*Porphyromonas gingivalis*	Primer	PG-For	CCGTTGAAAGACGGCCTAA	110 bp	[42]
		PG-Rev	CCCATCCCTTACCGGRA		
	Probe	PG-Pb	CCCGATGTGTTTCATTGACGGCATC		
*Tannarella forsythia*	Primer	TF-For	GAGGTTGTGGAAGGTATG	108 bp	[44]
		TF-Rev	GTAGATCAGAATGTACGGATT		
	Probe	TF-Pb	TCTCCGCTTATTTCGTGAC		

**Table 2 animals-13-01814-t002:** Statistical association (Cramer’s V) between different gravity (mild gingivitis, diastema, fracture, periodontal pocket, and others) of equine periodontal diseases found during clinical examination and positivity to *Treponema denticola*, *Tannerella forsythia* and *Porphyromonas gingivalis*, *Fusobacterium nucleatum*, *Veilonella parvula*, and *Prevotella intermedia*.

Variable	Categories	Lesion					Cramer’s V
		None	Fracture or Other	Diastema	Pocket	Total Number	
*Treponema denticola*	Negative	46	1	9	13	69	0.475
	Positive	14	13	8	21	56	
*Tannerella forsythia*	Negative	41	2	6	8	57	0.451
	Positive	19	12	11	26	68	
*Porphyromonas gingivalis*	Negative	60	151	17	31	122	0.257
	Positive	0	0	0	3	3	
*Fusobacterium nucleatum*	Negative	29	5	7	8	49	0.214
	Positive	31	9	10	26	76	
*Veilonella parvula*	Negative	6	1	0	2	9	0.13
	Positive	54	13	17	32	116	
*Prevotella intermedia*	Negative	47	8	11	21	87	0.187
	Positive	13	6	6	13	38	

**Table 3 animals-13-01814-t003:** Association between variables; region of animals’ origin, age, sex, breed, type of feed, BCS, bacterial positivity, and Equine Periodontal Disease (EPD).

Variable	Categories	Total Number	EPD (%)
Region of animal origin	Apulia	112	55 (49.1)
	Basilicata	2	2 (100)
	Campania	6	4 (66.6)
	Molise	5	4 (80.0)
Age	Juvenile horse (1–10 years)	41	13 (31.7)
	Adult horse (11–20 years)	58	33 (56.8)
	Elderly horse (>20 years)	26	19 (73.0)
Sex	Female	48	26 (54.1)
	Male	77	39 (50.6)
Breed	Murgese	6	2 (33.3)
	American Saddlebred	99	50 (50.5)
	Trotter	7	4 (57.14)
	Pony	13	9 (69.2)
Type of feed	Hay and concentrate	112	62 (55.3)
	Hay	3	3 (100)
BCS	BCS = 5	97	52 (53.6)
	1 ≤ BCS ≤ 4	9	8 (88.8)
	6 ≤ BCS ≤ 9	19	5 (26.3)
*Treponema denticola*	Negative	69	25 (36.2)
	Positive	56	40 (71.4)
*Tannerella forsythia*	Negative	57	16 (28.0)
	Positive	68	49 (72.0)
*Porphyromonas gingivalis*	Negative	122	63 (51.6)
	Positive	3	2 (66.6)
*Fusobacterium nucleatum*	Negative	49	21 (42.8)
	Positive	76	44 (57.8)
*Veilonella parvula*	Negative	11	4 (36.3)
	Positive	84	61 (72.6)
*Prevotella intermedia*	Negative	88	43 (48.8)
	Positive	37	22 (59.4)

**Table 4 animals-13-01814-t004:** Results of logistic regression analysis.

		B	S.E.	Wald	gl	Sign.	Exp(B)
First Phase	*Treponema denticola*	0.309	0.127	5.864	1	0.015	1.362
	*Tannerella forsythia*	−0.067	0.045	2.185	1	0.139	0.935
	*Fusobacterium nucleatum*	0.044	0.038	1.303	1	0.254	1.045
	*Porphyromonas gingivalis*	0.283	586.380	0.000	1	1.000	1.327
	*Prevotella intermedia*	0.044	0.045	0.918	1	0.338	1.045
	*Veilonella parvula*	−0.095	0.155	0.376	1	0.540	0.909
	Age	0.144	0.084	2.946	1	0.086	1.154
	Sex dicot (1)	0.238	1.150	0.043	1	0.836	1.269
	PonyYesNo (1)	−3.698	2.539	2.121	1	0.145	0.025
	Type of feed (1)	14.547	18,603.731	0.000	1	0.999	2,077,489.922
	Lesion	5.870	1.920	9.347	1	0.002	354.184
	Costant	−14.237	8.605	2.738	1	0.098	0.000

Variables entered in phase 1: *Treponema denticola*, *Tannerella forsythia*, *Fusobacterium nucleatum*, *Porphyromonas gingivalis*, *Prevotella intermedia*, *Veilonella parvula*, Age, Sex_dicot, PonyYesNo, Type of Feeding, Lesion.

## Data Availability

Not applicable.

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
