# Peer review of "Bacterial Periodontitis in Horses: An Epidemiological Study in Southern Italy"

_animals, 2023, doi:10.3390/ani13111814_

Round 1
Reviewer 1 Report
Thank you very much for the manuscript on the prevalence of certain bacteria species in the mouth of horses with different dental status in southern Italy. Such epidemiological studies are very important for a better understanding of the connections between different risk factors and the occurrence of EPD as a multifactorial disease.
In my point of view there are a few points to revise and reconsider:
1. Line 17 ff.: Please rewrite so that the sentence becomes more comprehensible. Since the authors only examined the incidence of Tannerella and Treponema spp. belonging to the RCB in my view it is misleading to write in one sentence something about the proportion of horses where these species were detected and the low amount of all examined bacteria belonging to the RCB. Perhaps one could but this into two separate sentences: “Tannerella spp. and Treponema spp. were detected with a significantly higher proportion in horses affected by EPD than in healthy animals. Overall, examined pathogens belonging to RCB were detected in low amounts.”
2. Same lines: What is meant by low amounts? Is it the proportion of horses with examined RCB besides Tannerella spp. and Treponema spp. or the number of nucleic acids in the samples? If it is the number of nucleic acids (bacteria) in the sample, can you specify this?
3. Line 34 f.: How was this staging done or is the type of lesion (pocket, diastema…) meant by severity or is there a staging like multiple pockets, diastemata?
4. Line 32 ff.: Please see 1. and 2.
5. Line 57: Please correct the term “envirolnmental”
6. Line 118 f.: Was only an assessment of BCSs done to evaluate the health status or more (general examination, questions about comorbidity)?
7. Line 123: Please insert spaces before SI units and check this unit (Is mg/kg meant?)
8. Line 127 ff.: From which teeth or tooth were the two oral swabs taken in animals with healthy dental status? Can you please insert this with one sentence?
9. Line 133: Please insert spaces before SI units (+4 °C) and check this for the rest of the manuscript.
10. Figure 1 point 2 and line 138: Were the samples centrifuged by 1000 g or 1,000 g?
11. Figure 1 point 3: Please insert spaces before ml and °C.
12. Line 145: Please use the correct abbreviation mpPCR.
13. Table 1: From my point of view it would be better to write primers instead of nucleotide.
14. Line 202 and table 3: In line 202 it is written that 96/125 examined horses were American Saddlebred horses and in table 3 99/125 horses were Sella Italiano.
15. Table 3: It is very interesting to examine the type of feed. Are there more detailed analyzes concerning this like how the horses were fed (for example hay nets, which hay was fed, ad libitum,…)?
16. Table 4: Please insert leading zeros.
17. Line 252: The space between the words seems to be unequal. Can you please check this for the document?
18. Line 280: Please correct the literature references “[22[1846]”
19. Line 287 ff.: Indeed, the factors are interesting, but also some secondary diseases may play an important role. Did you any analyzes concerning secondary diseases of the examined horses for example PPID, EOTRH? Many horses seem to be clinically healthy but suffer from EOTRH.
20. Line 301ff.: It would be very interesting to collect some data concerning the oral routine of examined horses.
21. Line 365 ff.: Please rewrite the sentence “should be better informed and able to detect”
Author Response
Dear Reviewer,
thank you for giving us the opportunity to submit a revised draft of our manuscript titled: “Bacterial Periodontitis in horses: an epidemiological study in southern Italy”, Manuscript ID: animals-2340910, to Animals.
We appreciate the time and effort that you have dedicated to providing your valuable feedback on our manuscript. We are grateful for your insightful comments on our paper. We have been able to incorporate changes to reflect the suggestions provided by the editor and reviewers. We have highlighted the changes within the manuscript.
Here is a point-by-point response to your comments and concerns.
Comments from Reviewer 1 (R1)
Thank you very much for the manuscript on the prevalence of certain bacteria species in the mouth of horses with different dental status in southern Italy. Such epidemiological studies are very important for a better understanding of the connections between different risk factors and the occurrence of EPD as a multifactorial disease.
In my point of view there are a few points to revise and reconsider:
R1.1: Line 17 ff.: Please rewrite so that the sentence becomes more comprehensible. Since the authors only examined the incidence of Tannerella and Treponema spp. belonging to the RCB in my view it is misleading to write in one sentence something about the proportion of horses where these species were detected and the low amount of all examined bacteria belonging to the RCB. Perhaps one could but this into two separate sentences: “Tannerella spp. and Treponema spp. were detected with a significantly higher proportion in horses affected by EPD than in healthy animals. Overall, examined pathogens belonging to RCB were detected in low amounts.”
Reply to R1.1: We agree with the referee and have rephrased the sentences according to the kindly suggestions of the reviewer.
R1.2: Same lines: What is meant by low amounts? Is it the proportion of horses with examined RCB besides Tannerella spp. and Treponema spp. or the number of nucleic acids in the samples? If it is the number of nucleic acids (bacteria) in the sample, can you specify this?
Reply to R1.2: In line 19 in “low amounts” refers to the low percentage of animals positive to pathogens belonging to RCB. We agree with the referee and to avoid misunderstanding we have rephrased the sentence.
R1.3: Line 34 f.: How was this staging done or is the type of lesion (pocket, diastema…) meant by severity or is there a staging like multiple pockets, diastemata?
Reply to 1.3: As reported by Gienchi in 2010 and Kennedy and Dixon in 2018, a different score was assigned depending on the clinical severity of the EPD identified during the oral clinical examination. That is, a score from 0 to 4 was assigned, where 0 indicated the absence of a lesion, 1 indicated the presence of a minor lesion, i.e. gingivitis, 2 indicated early periodontal disease, i.e. diastema, 3 indicated moderate periodontal disease represented by fractures, 4 indicated advanced periodontal disease characterized by pocket.
This categorization of the variable "type of lesion" was reported in the section “Data analyses” in the paragraph "Material and methods", lines 194-196: "The EPD evaluated at the oral examination was classified into different categories (mild gingivitis, diastema, fracture, pocket, and others) according to the severity of the disease, as previously suggested [9,44]."
R1.4: Line 32 ff.: Please see 1. and 2.
Reply to R1.4: Please see the reply to R1.1 and R1.2. We also incorporated in the revised manuscript the kindly suggestion of the reviewer.
R1.5: Line 57: Please correct the term “envirolnmental”
Reply R1.5: We agree with the referee and have corrected the typo.
R1.6: Line 118 f.: Was only an assessment of BCSs done to evaluate the health status or more (general examination, questions about comorbidity)?
Reply R1.6: To evaluate the general state of health of each patient, only the BCS was evaluated, since the dental vet was called by another practitioner or by the owners itself for a specific oral examination. The BCS is a crucial health parameter for the dental vet who is preparing to visit a horse. In fact, a horse with an oral pathology will hardly eat, and the loss of weight, as well as competitive performance, is the first clinical sign that the owner reports during the anamnesis. During the oral evaluation, the practitioner summarily evaluated the general health condition of each patient, concentrating on the evaluation of the mouth.
R1.7: Line 123: Please insert spaces before SI units and check this unit (Is mg/kg meant?)
Reply to R1.7: In line 123 with this wording “10-30 mcg/Kg of detomidine hydrochloride” we means the dosation of detomidine hydrochloride that was subminstraid in 10-30 millionth of a gram (mcg) per Kilogram (kg) of animal weith. This type of diction is really very common when you want to indicate the dosage of the drugs used, in particular for anesthetics.
R1.8: Line 127 ff.: From which teeth or tooth were the two oral swabs taken in animals with healthy dental status? Can you please insert this with one sentence?
Reply R1.8: to our knowledge, the teeth that mostly are interested by pathology in the equine species are cheek theeth. During the dental clinic visit, the clinician evaluated the whole mouth and after having established the absence of periodontal disease he carried out sampling from the cheek theeth. We agree with the suggestion of the reviewer and we have incorporated this information throughout the revised manuscript.
R1.9: Line 133: Please insert spaces before SI units (+4 °C) and check this for the rest of the manuscript.
Reply to R1.9: We agree with the referee and have incorporated his suggestion throughout the revised manuscript and the figure 1.
R1.10: Figure 1 point 2 and line 138: Were the samples centrifuged by 1000 g or 1,000 g?
Reply to R1.10: We agree with the referee and have incorporated his suggestion throughout the revised manuscript and the figure 1, now there is more consistency.
R1.11: Figure 1 point 3: Please insert spaces before ml and °C.
Reply to R1.11: We agree with the referee and have incorporated his suggestion throughout the revised figure 1.
R1.12: Line 145: Please use the correct abbreviation mpPCR.
Reply to R1.12: We agree with the referee, and to avoid misunderstanding our protocol, which is a multiplex real-time PCR (or multiplex qPCR), we prefer this kind of abbreviation, that was inserted throughout the revised manuscript.
R1.13: Table 1: From my point of view, it would be better to write primers instead of nucleotide.
Reply to R1.13: We agreed with the referee and wrote “primers” instead of “nucleotides”.
R1.14: Line 202 and table 3: In line 202 it is written that 96/125 examined horses were American Saddlebred horses and in table 3 99/125 horses were Sella Italiano.
Reply to R1.13: We agreed with the referee, and the data was corrected after a careful check.
R1.15: Table 3: It is very interesting to examine the type of feed. Are there more detailed analyzes concerning this like how the horses were fed (for example hay nets, which hay was fed, ad libitum,…)?
Reply R1.15: We agree with the reviewer's comment, given that the type of feed provided to the horse would appear to be a predisposing factor for EPD. Unfortunately, in southern Italy, where the study was conducted, horse nutrition is traditionally linked to the climate. For this reason, the 97.3% of examined horses are mainly fed with oat hay supplemented with concentrate consisting of a mix of grains, with an administration of 3 times a day. No further information was reported by the practitioner.
R1.16: Table 4: Please insert leading zeros.
Reply R1.16: We agree with the referee and have incorporated his suggestion throughout the revised table 4.
R1.17: Line 252: The space between the words seems to be unequal. Can you please check this for the document?
Reply R1.17: We agree with the referee and have carefully checked in the revised version of the manuscript.
R1.18: Line 280: Please correct the literature references “[22[1846]”
Reply to R1.18: We agree with the referee and have carefully checked and correct the literature references.
R1.19: Line 287 ff.: Indeed, the factors are interesting, but also some secondary diseases may play an important role. Did you any analyzes concerning secondary diseases of the examined horses for example PPID, EOTRH? Many horses seem to be clinically healthy but suffer from EOTRH.
Reply R1.19: All data reported by the practioner during the dental visit were inserted in the manuscript. Secondary disorders such as PPID, EOTRH, were not reported as they were not clinically detected. Undoubtedly, further studies should be carried out to provide more information on the epidemiology, the pathogenesis, and risk factors such as the quality and modalities of the food supplied, and the association with secondary dental pathologies.
R1.20: Line 301ff.: It would be very interesting to collect some data concerning the oral routine of examined horses.
Reply R1.20: None of the horses screened had a regular oral routine, so we did not consider it appropriate to include this information in the statistical analysis. However we have included this information in the sampling collection section of the materials and methods lines: 161,162.
R1.21: Line 365 ff.: Please rewrite the sentence “should be better informed and able to detect” to the "Discussion."
Reply R1.21: We agree with the referee and have rephrased the sentence.

Reviewer 2 Report
The manuscript describes a study of EPD and five human associated bacteria using RT-qPCR. Describes the risk factors to disease. Tannerella spp and Treponema spp were detected at a significantly high proportion in horses with EPD with increasing age.
Suggestions or comments to Authors:
Abstract: Line 24-26 - suggest rewrite to indicate that EPD is initiate by specific bacterial species and are not the secondary cause of disease
Introduction: line 48-49. Where is the evidence that mechanical abrasions cause EPD. Again, it is not an infection but a chronic inflammatory disease due to the host response to the specific periodontopathic bacteria.
Line 82 …bacteria are not found as “pure” but in mixed growths.
Would be useful to readers in the introduction to mention that equine dentition has hypsodontal dentition of cheek teeth
Line 94 …would be useful to include Ref 10 Kennedy et al 2016
Lines 93-96 …RCB are human isolates especially Porphyromonas gingivalis while Porphyromonas gulae is the animal specific isolate…While Pg is found in low levels in human PD it can act as a keystone pathogen. Would be good if mention of specific plaque hypothesis, keystone pathogen of even dysbiosis as discussed in Kennedy and Dixon 2018. Cite reference to this. Suggest re-read this paper.
In the discussion synergistic relationships are not only involved with inflammatory responses but in the development of the oral biofilm such as the bridging of Fusobacterium nucleatum
Suggest spell checker
Author Response
Dear Reviewer,
thank you for giving us the opportunity to submit a revised draft of our manuscript titled: “Bacterial Periodontitis in horses: an epidemiological study in southern Italy”, Manuscript ID: animals-2340910, to Animals.
We appreciate the time and effort that you have dedicated to providing your valuable feedback on our manuscript. We are grateful for your insightful comments on our paper. We have been able to incorporate changes to reflect the suggestions provided by the editor and reviewers. We have highlighted the changes within the manuscript.
Here is a point-by-point response to your comments and concerns.
Comments from Reviewer 2 (R2)
The manuscript describes a study of EPD and five human associated bacteria using RT-qPCR. Describes the risk factors to disease. Tannerella spp and Treponema spp were detected at a significantly high proportion in horses with EPD with increasing age.
Suggestions or comments to Authors:
R2.1: Abstract: Line 24-26 - suggest rewrite to indicate that EPD is initiated by specific bacterial species and are not the secondary cause of disease.
Reply R2.1: We agree with the referee and have rephrased the sentences according to the kindly suggestions of the reviewer.
R2.2: Introduction: line 48-49. Where is the evidence that mechanical abrasions cause EPD. Again, it is not an infection but a chronic inflammatory disease due to the host response to the specific periodontopathic bacteria.
Reply R2.2: We agree with the referee and have modified the revised manuscript according to the suggestion.
R2.3: Line 82 …bacteria are not found as “pure” but in mixed growths.
Reply R2.3: We agree with the referee and have specified this information in the revised manuscript.
R2.4: Would be useful to readers in the introduction to mention that equine dentition has hypsodontal dentition of cheek teeth
Reply R2.4: We agree with the referee and have incorporated his suggestion throughout the revised manuscript.
R2.5: Line 94 …would be useful to include Ref 10 Kennedy et al 2016.
Reply R2.5: We agree with the referee and have include Ref 10 Kennedy et al 2016 in the revised manuscript.
R2.6: Lines 93-96 …RCB are human isolates especially Porphyromonas gingivalis while Porphyromonas gulae is the animal specific isolate…While Pg is found in low levels in human PD it can act as a keystone pathogen. Would be good if mention of specific plaque hypothesis, keystone pathogen of even dysbiosis as discussed in Kennedy and Dixon 2018. Cite reference to this. Suggest re-read this paper.
Reply 2.6: We agree with the referee and have incorporated his suggestion throughout the revised manuscript.
R2.7: In the discussion synergistic relationships are not only involved with inflammatory responses but in the development of the oral biofilm such as the bridging of Fusobacterium nucleatum
Reply 2.7: We agree with the referee and have incorporated his suggestion throughout the revised manuscript.
R2.8: Suggest spell checker
Reply R2.8: We are grateful to the reviewer for the suggestion, the spelling in the revised manuscript has been carefully checked.

Round 2
Reviewer 1 Report
Thank you very much for the revised version. The proposed changes are well included.